# A Single Episode of Cortical Spreading Depolarization Increases mRNA Levels of Proinflammatory Cytokines, Calcitonin Gene-Related Peptide and Pannexin-1 Channels in the Cerebral Cortex

**DOI:** 10.3390/ijms24010085

**Published:** 2022-12-21

**Authors:** Maria N. Volobueva, Elena M. Suleymanova, Maria P. Smirnova, Alexey P. Bolshakov, Lyudmila V. Vinogradova

**Affiliations:** 1Department of Molecular Neurobiology, Institute of Higher Nervous Activity and Neurophysiology, Russian Academy of Sciences, Butlerova Street 5A, 117485 Moscow, Russia; 2Department of Conditioned Reflexes and Physiology of Emotion, Institute of Higher Nervous Activity and Neurophysiology, Russian Academy of Sciences, Butlerova Street 5A, 117485 Moscow, Russia

**Keywords:** cortical spreading depolarization, neuroinflammation, interleukin-1beta, tumor necrosis factor, interleukin-6, calcitonin gene-related peptide, pannexin-1 channel, migraine aura

## Abstract

Cortical spreading depolarization (CSD) is the neuronal correlate of migraine aura and the reliable consequence of acute brain injury. The role of CSD in triggering headaches that follow migraine aura and brain injury remains to be uncertain. We examined whether a single CSD occurring in awake animals modified the expression of proinflammatory cytokines (Il1b, TNF, and Il6) and endogenous mediators of nociception/neuroinflammation-pannexin 1 (Panx1) channel and calcitonin gene-related peptide (CGRP), transforming growth factor beta (TGFb) in the cortex. Unilateral microinjury of the somatosensory cortex triggering a single CSD was produced in awake Wistar rats. Three hours later, tissue samples from the lesioned cortex, intact ipsilesional cortex invaded by CSD, and homologous areas of the contralateral sham-treated cortex were harvested and analyzed using qPCR. Three hours post-injury, intact CSD-exposed cortexes increased TNF, Il1b, Panx1, and CGRP mRNA levels. The strongest upregulation of proinflammatory cytokines was observed at the injury site, while CGRP and Panx1 were upregulated more strongly in the intact cortexes invaded by CSD. A single CSD is sufficient to produce low-grade parenchymal neuroinflammation with simultaneous overexpression of Panx1 and CGRP. The CSD-induced molecular changes may contribute to pathogenic mechanisms of migraine pain and post-injury headache.

## 1. Introduction

The interplay between immune and neuronal systems is involved in the regulation of physiological functions of the brain and pathogenic mechanisms of many neurological conditions. Innate immune cells of brain parenchyma activated by exogeneous or endogenous stress signals initiate an orchestrated neuroinflammatory response with enhanced production of cytokines [1,2]. The immune response of the brain may be triggered by enhanced neuronal activity in the absence of significant pathological conditions (neurogenic neuroinflammation) [2]. Neuroinflammation is known to produce both beneficial (tissue remodeling, repair, coping with enhanced metabolic demands, etc.) and detrimental (pain promoting, sensitization, neurotoxicity, etc.) effects on brain function [1].

Cortical spreading depolarization (CSD), a wave of transient neuroglial depolarization, is implicated in the pathophysiology of migraine and acute brain injury. CSD is accepted to underlie migraine aura–neurological symptoms preceding headache in a subset of migraine patients. However, the role of CSD in the mechanisms of intracranial headache remains to be uncertain. Migraines have a strong genetic component, and genetic factors are likely to play a role in individual susceptibility to CSD [3]. Mounting evidence indicates the critical role of the calcitonin gene-related peptide (CGRP, encoded by gene *Calca*) and neuroinflammation in migraine pathogenesis [4,5,6,7]. Cortical and parameningeal inflammatory activity, as well as increased serum levels of CGRP and proinflammatory cytokines, have been reported during migraine attacks in patients [8,9,10]. Preclinical studies have shown that CSD also stimulates CGRP release and production of proinflammatory markers in the cerebral cortex [5,11,12,13,14] and activates trigeminovascular nociception [15]. CSD has been demonstrated to trigger parenchymal inflammatory cascades via rapid opening of large-pore mechanosensitive pannexin 1 (Panx1) channels [14].

The experimental evidence for the proinflammatory effect of CSD has been obtained in anesthetized rodents, and most of the effects were observed following multiple CSDs. However, migraine is mostly associated with a single aura (CSD) developing in non-anesthetized individuals. General anesthesia can modify an innate immune response of the injured brain [16,17] and produce long-term cognitive impairments [18]. Patterns and effects of CSD are known to depend on the baseline functional state of the brain. Significant changes in the state under general anesthesia suggest that characteristics of neuroinflammatory responses may differ in awake and anesthetized animals. In the present study, we examined whether a single CSD induced in undrugged freely moving animals modifies the expression of key proinflammatory markers (Il1b, TNF, and Il6) and endogenous mediators of nociceptive transmission and neuroinflammation (CGRP, Panx1 channels and transforming growth factor beta) in the cerebral cortex. We hypothesized that a single CSD, underlying migraine aura symptoms, enhances production of the proinflammatory cytokines and the mediators playing an important role in headache pathogenesis. The transcriptional changes were assessed at three hours after CSD induction. Given that migraine headache typically develops within one hour after aura and lasts 7–48 h, the time point corresponds to the post-aura headache period of a migraine attack.

CSD-related neuroinflammation may be also relevant to other pathological conditions in which CSD is involved. CSD reliably occurs in the injured human brain [19] and injury-induced CSD can be involved in pathogenesis of post-injury headaches commonly having migraine-like character [20,21]. Both brain injury and CSD are known to activate programs of acute inflammatory gene expression in brain parenchyma [1,12,22,23]. Moreover, local chemical or mechanical stimulation of the cortex usually used for induction of CSD in experimental animals can produce local tissue damage and inflammation overlapping with proinflammatory effects of CSD itself. To dissect the transcriptional responses induced by CSD and acute injury, we compared expression levels of the targeted genes in the injured and intact cortical tissue following a CSD episode. It was found that strong neuroinflammatory response with upregulation of all key proinflammatory cytokines is observed in the injured cortex, while mild mRNA overexpression of CGRP and immune-relevant molecules follows a single CSD episode in the intact non-injured cortex.

## 2. Materials and Methods

### 2.1. Subjects

Twenty-two male Wistar rats (300–350 g, Scientific center for Biomedical Technologies of the Federal Medical and Biological Agency, Russia) were housed in a temperature-controlled vivarium (22 °C ± 2 °C, a 12-h light/dark cycle, lights on at 08.00 h) with food and water ad libitum. Animals were kept 5 per cage before surgery and individually after surgery. All experimental procedures were conducted in accordance with the ARRIVE guidelines and the Directive 2010/63/EU for animal experiments. The study protocol was approved by the Ethics Committee of the Institute of Higher Nervous Activity and Neurophysiology of the Russian Academy of Sciences. Every effort was made to minimize animal suffering and to ensure reliability of the results.

### 2.2. Stereotaxic Surgery

Under chloral hydrate anesthesia (400 mg/kg, i.p. AppliChem, Darmstadt, Germany), rats were implanted with the bilateral stainless-steel guide cannulas (0.6 mm outer diameter and 0.3 mm inner diameter) aimed at the primary somatosensory cortex (AP: −2.8, ML: 4.8, and DV: 1.5) [24]. Twelve rats used for electrophysiological experiments were also equipped with recording electrodes (silver, diameter of 0.3 mm) implanted in the frontal cortex (AP: +1.2, ML: ± 2.3, and DV: −1.8). Ten rats used for qPCR analysis were equipped with identical bilateral guide cannulas implanted in the same regions of the primary somatosensory cortex, as in rats used in electrophysiological experiments (one rat from the qPCR group was discarded from analysis due to technical problems). The guide cannulas and electrodes were fixed on the skull with acrylic dental plastic. A stylus of the same length as guide cannula was inserted into it to prevent clogging. Experiments started two weeks post-surgery. All animals were prehandled and habituated to the stylus removal daily at a week before the start of experiments. 

### 2.3. Experimental Design

In electrophysiological experiments, freely moving rats were individually placed in a shielded experimental chamber (60 × 40 × 40 cm), and the implanted connector was attached to the recording cable. After a 5 min habituation period and a 10 min baseline (preinjury) recording of wideband cortical activity (0–100 Hz, 1 kHz sampling rate), a unilateral controlled microinjury of the somatosensory cortex was induced. During the microinjury induction, an awake rat was gently handled, and the injection cannula (0.3 mm diameter) was inserted, extending 1.0 mm from the tip of the guide cannula. Cortical activity was recorded for 10 min post-injury using a four-channel, high-input impedance (1 gΩ) DC amplifier and a/d converter (E14-440, L-Card, Moscow, Russia) with simultaneous video-monitoring of behavior. SD occurrence was identified based on detection of a characteristic abruptly developing high-amplitude (larger 2 mV) negative shift of the slow potential (Figure 1). Sham injury was produced by insertion of a short injection cannula not extending from the guide cannula into cortical parenchyma. 

In PCR experiments, freely moving rats were individually placed in an experimental chamber (40 × 40 × 40 cm), and their behavior was video recorded during all the experiment. After a 5 min habituation period, a sham injury of the right somatosensory cortex was produced by the abovementioned method (insertion of a short cannula not damaging neuronal tissue) and, five minutes later, a standard microinjury of the left somatosensory cortex was induced by the way described previously (1 mm protruding of an injection cannula beyond the tip of an implanted cannula guide into cortical parenchyma). Three hours after the injury and CSD induction, the animals were decapitated and tissue samples from the perilesional left somatosensory cortex, sham-treated right somatosensory cortex, and right and left intact frontal cortical regions were extracted and stored at −80° C for further analysis (Figure 2). 

### 2.4. RNA Isolation, Reverse Transcription, and qPCR

To isolate total RNA from tissues, the samples were homogenized with ExtractRNA reagent (Evrogen, Moscow, Russia) according to the protocol provided by the manufacturer. Before the reverse transcription, residual DNA in samples was removed by treatment with DNase I (1U/µL) (Thermo Scientific, Waltham, MA, USA) according to the protocol specified by the manufacturer; then, the reverse transcription was carried out using MMLV revertase (Evrogen, Russia), and murine RNase Inhibitor (New England Biolabs, Ipswich, MA, USA) was used to prevent possible degradation of RNA. cDNA samples were stored at −20 °C.

Relative mRNA quantities of each gene were evaluated by real-time PCR (rtPCR) using a 5X qPCRmix-HS SYBR + LowROX reaction mixture (Evrogen, Russia), according to the protocol specified by the manufacturer, on a Biorad CFX 384 Real Time PCR. Relative quantities of mRNAs for genes expressed in the regions of interest were normalized to the geometric mean of the mRNA expression levels for the Ywhaz, Osbp, and Hprt1 housekeeping genes. The negative control PCR was with the products of DNase I treatment. Gene expression was analyzed by the 2^−ΔΔCt^ method. The sequences of primers used are shown in Table 1.

### 2.5. Statistical Analysis

Statistical analysis was performed using Statistica software 8.0 (StatSoft). Wilks–Shapiro test showed non-normal distribution of the data; therefore, non-parametric Friedman ANOVA test was used to analyze expression levels of the targeted genes. Post hoc non-parametric Wilcoxon test with Bonferroni correction for multiple comparisons was used to assess differences in mRNA levels between four cortical regions—perilesional somatosensory cortex, intact ipsilesional frontal cortex, and homologous areas of the opposite hemisphere. The significance level was set at *p* < 0.017.

## 3. Results

Electrophysiological experiments with DC recording of cortical activity showed that a controlled microinjury of the somatosensory cortex reliably induced a single ipsilateral CSD in awake rats (Figure 1). The CSD was recorded following 96% (23/24) of the cortical injures. CSD spread unilaterally across one hemisphere from the injury site. Electrographic recordings showed that CSD always invaded the frontal cortex of the ipsilesional hemisphere and never propagated to the contralateral cortex (Figure 1). Sham lesions did not produce CSD (0/24). 

In freely behaving rats used for subsequent qPCR analysis, an acute controlled microinjury caused a focal lesion of the somatosensory cortex, identical to that produced in electrophysiological experiments. Friedman ANOVA showed highly significant changes in cortical mRNA expression of all targeted genes—Il1b (*p* = 0.0007), TNF (*p* = 0.00002), Il6 (*p* = 0.0015), TGFb (*p* = 0.0098), Panx1 (*p* = 0.0001), and CGRP (*p* = 0.0039)—at three hours after the injury. Post hoc comparison using the Wilcoxon test with Bonferroni correction showed that all the genes, except CGRP, were upregulated at the perilesional cortex compared with the homologous sham-treated (intact) cortex of the contralateral hemisphere. Some of the molecules were also upregulated in the ipsilesional intact cortex invaded by a single CSD compared with the contralateral sham-treated frontal cortex (Figure 3 and Figure 4). 

At three hours post-injury, the cortical parenchyma near the injury site strongly increased expression in all proinflammatory cytokines Il1b (47-fold), TNF (20-fold), and Il6 (20-fold) (Figure 3A). In the intact ipsilesional cortex that experienced a single CSD episode, the cytokines were also upregulated but in less degree—Il1b and TNF showed a twofold and sixfold increases, respectively, Il6 did not change its expression (*p* = 0.314) (Figure 3B). In the lesioned hemisphere, mRNA expression levels of the proinflammatory cytokines Il1b, TNF, and Il6 were significantly higher in the injured site than in the distal uninjured cortex invaded by a single CSD (*p* < 0.0085 for all the genes). No regional difference was found in the contralesional sham-treated hemisphere. That is, injured cortical tissue exhibited an intense neuroinflammatory response, whereas the uninjured CSD-exposed cortex showed low-grade neuroinflammation. Il1b was maximally upregulated in the perilesional cortex, while TNF was the most affected proinflammatory cytokine in the intact cortex that experienced a single CSD (Figure 3).

Transforming growth factor beta and Panx1 channels showed low but significant upregulation in the perilesioned cortex—2-fold and 1.4-fold increases, respectively—compared with the contralateral sham-treated cortex (Figure 4A). Unexpectedly, CGRP mRNA showed no significant changes near the site of the injury compared with the contralateral cortical region. The intact cortex of the ipsilesional hemisphere invaded by a single CSD slightly but significantly increased Panx1 (*p* < 0.0085) and CGRP (*p* < 0.017) mRNA levels and tended to upregulate TGFb (*p* = 0.028) (Figure 4B). Comparison of injured and intact tissues within the hemispheres revealed the only significant difference—in the lesioned hemisphere Panx1 expression was higher in the intact CSD-experienced cortex than near the injury site (*p* < 0.0085). That is, in the intact cortex, a single CSD episode upregulated Panx1 and CGRP for at least three hours, while in the injured cortex, CSD increased expression of Panx1 and TGFb. 

## 4. Discussion

This paper shows for the first time that a single CSD developing in the intact cerebral cortex of awake animals is sufficient to produce parenchymal neuroinflammation and upregulation of Panx1 and CGRP at three hours post-CSD. The perilesional tissue showed strong local inflammatory response with increased mRNA expression of all key proinflammatory cytokines Il1b, TNF, and Il6. The response reflects the well-known rapid activation of an innate immune response evoked by an acute injury rather than the CSD-associated tissue depolarization. The pronounced upregulation of proinflammatory signaling cascades at the lesioned cortex differed from the relatively mild neurogenic neuroinflammation induced by a single CSD in the injury-remote intact cortex where TNF and Il1b, but not Il6, increased their expression 3 h later. This finding is in line with the data obtained in anesthetized mice and showed upregulation of Il1b and TNF without changes in Il6 at 1–4 h after a single optogenetically induced CSD [12]. Increased mRNA expression of TNF and Il1b has also been reported at 2–4 h after multiple CSDs [11,12,13,25]. In the current study, overexpression of Il6 and TGFb was restricted to the region surrounding the primary injury site which points to the possibility that CSD plays different roles in the development of neuroinflammation in injured and intact neuronal tissue. 

Most CSD-associated events (breakdown of transmembrane ion gradients, glutamate release, cellular depolarization, etc.) last minutes in well-nourished neuronal tissue, but the relatively brief and reversible dysfunction is followed by sustained upregulation of proinflammatory cytokines and nociceptive mediators in the cerebral cortex. Given the timing of the aura phase relative to migraine headache, the activation of the parenchymal inflammatory and nociceptive pathways for hours following a single CSD episode supports a potential involvement of CSD in mechanisms of pain generation in migraine with aura. The present study shows that TNF is the most affected proinflammatory cytokine 3 h after a single CSD and suggests a close link between the cytokine and CSD. The stronger upregulation of TNF compared with Il1b at 4 h post-CSD has also been found in other experimental studies [11,12]. Clinical evidence for enhanced TNF plasma levels during migraine attacks in patients [26] support an important role of the cytokine in the pathogenesis of migraines. Given that TNF promotes mechanical sensitization of meningeal nociceptors [27] and CGRP release in trigeminal ganglion [28], CSD-induced upregulation of TNF may contribute to mechanisms underlying migraine headache. However, the recently described suppressive effect of TNF on CSD [29] suggests that the post-CSD upregulation of TNF may have a neuroprotective function, too. 

The current study, for the first time, shows persistent overexpression of Pannexin-1 (Panx1) mRNA following a single CSD. The fact that Panx1 was more highly upregulated in intact cortical tissue that experienced a CSD episode than in the injured cortex suggests that CSD is the main driver of the increase in Panx1 expression. Pannexin-1, a large-pore membrane channel, is a potential therapeutic target for many clinical conditions and a key regulator of post-injury and post-CSD inflammatory responses [14,30]. CSD (even a single episode) has been shown to activate parenchymal inflammatory cascades and promote sustained activation of trigeminal afferents via opening of neuronal Panx1 channel [14]. It has been demonstrated that rapid activation of Panx1 within minutes following CSD triggers inflammasome formation and the release of proinflammatory molecules such as Il1b. Inhibitors of Panx1 channels suppress the inflammatory signaling cascade and abolish CSD-induced trigeminal activation [14]. Growing evidence supports the role of pathological activity of Panx1 channels in migraine headache and mechanisms of central sensitization during persistent pain [31,32]. Therefore, sustained upregulation of Panx1 channels following a single CSD event found in our study may lead to intensification of cytokine production, aggravation of neuroinflammatory response, and accumulation of cellular stress with successive CSD episodes that may contribute to the sensitization process and migraine chronification. 

Furthermore, we showed that a single CSD enhances CGRP mRNA expression in the intact cortical parenchyma three hours later. Previously, it was demonstrated that multiple, but not a single, CSDs induced in anesthetized rats increased levels of CGRP mRNA and peptide in the ipsilateral cortex 24 h later [33]. Our data about a lack of significant changes in CGRP expression in the perilesioned cortical tissue indicate that CSD is a more important factor for CGRP upregulation than acute parenchymal injury. CGRP is known to play a critical role in nociceptive transmission and mechanisms of pain generation in migraine: CGRP serum levels elevate early in the migraine attacks [5,34], exogeneous CGRP triggers headache in migraine patients, and successful treatment of headaches leads to normalization of CGRP levels [4,34]. The relationship between increased serum levels of CGRP and proinflammatory cytokines has been reported in migraine patients [10]. Based on the evidence, it can be suggested that excess of CGRP in the cortex following a single CSD may play a role in the generation of migraine headache. 

Given the regular occurrence of CSD in the injured human brain, CSD may be an important factor influencing injury-related molecular changes in the distal intact regions of the cortex and contributing to pathophysiological mechanisms of post-traumatic migraine-like headaches. Since CSD occurs not only in severe brain insults [19] but also in association with seizures and small injuries of neuronal tissue [35,36,37,38], the CSD-induced molecular changes described in the present study could be involved in the generation of intracranial pain following neurosurgery (post-operative headaches) and epileptic seizures (post-ictal headaches). 

In conclusion, the present study showed that brief intense neuroglial depolarization during a single CSD episode is sufficient to trigger a low-grade neuroinflammatory response in the intact cortical tissue. Being a neuronal correlate of migraine aura and a reliable consequence of acute brain insults, CSD may be an important player in pathogenic mechanisms of post-aura and post-injury headaches by enhancing the production of proinflammatory cytokines and pain-mediating substrates, including CGRP. The mechanisms underlying CSD-induced upregulation of gene expression may involve calcium signaling [39] activated by the calcium influx into the cytoplasm during CSD, although other mechanisms cannot be excluded. The present study was limited by analysis of mRNA levels, and further investigation of CSD effects on protein synthesis is needed.

## Figures and Tables

**Figure 1 ijms-24-00085-f001:**
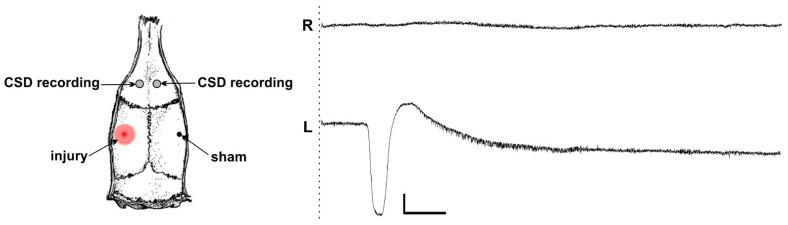
A single unilateral cortical spreading depolarization (CSD) induced by a local parenchymal injury of the somatosensory cortex and its propagation to the ipsilesional frontal cortex of an awake rat. Scheme shows the localization of the cortical microinjury, sham stimulation, and recording electrodes. A representative trace shows a single injury-induced CSD propagated to the left frontal cortex (L) but not invaded the contralesional right frontal cortex (R). The time of microinjury is marked by dashed line. Scale bars—50 s and 2 mV. Negativity is directed downward.

**Figure 2 ijms-24-00085-f002:**
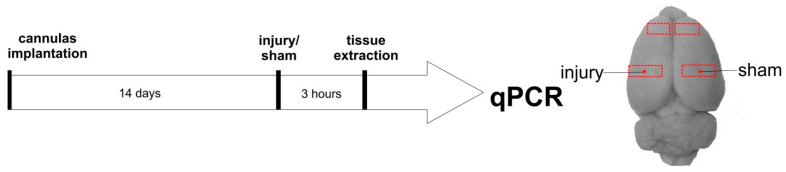
Experimental design and cortical regions harvested for qPCR analysis.

**Figure 3 ijms-24-00085-f003:**
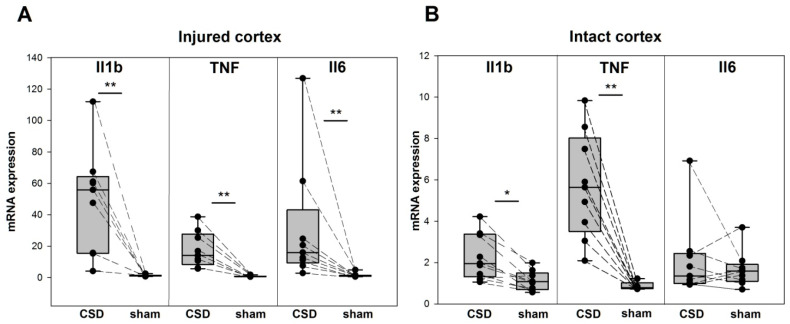
Proinflammatory cytokine expression following a single CSD in the injured and intact regions of the cerebral cortex. Levels of Il1b, TNF, and Il6 mRNA (medians with interquartile ranges and individual data from each rat, *n* = 9) in the perilesional tissue of the somatosensory cortex (**A**) and in the intact tissue of the frontal cortex (**B**) of the lesioned (CSD) and opposite (sham) hemispheres. * *p* < 0.017 and ** *p* < 0.0085—ipsilesional versus sham-treated cortical regions (post hoc Wilcoxon test with Bonferroni correction).

**Figure 4 ijms-24-00085-f004:**
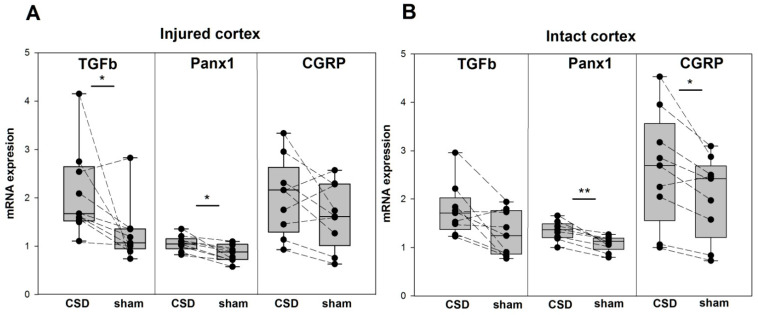
Expression of TGFb, Panx1, and CGRP following a single CSD in the injured and intact regions of the cerebral cortex. Levels of TGFb, Panx1, and CGRP mRNA (medians with interquartile ranges and individual data from each rat, *n* = 9) in the perilesional tissue of the somatosensory cortex (**A**) and in the intact tissue of the frontal cortex (**B**) of the lesioned (CSD) and opposite (sham) hemisphere. * *p* < 0.017 and ** *p* < 0.0085—ipsilesional versus sham-treated cortical regions (post hoc Wilcoxon test with Bonferroni correction).

**Table 1 ijms-24-00085-t001:** Sequences of primers and annealing temperature used for each primer pair (5′ > 3′).

Gene	Forward	Reverse	T_anneling_, °C
Calca	AGT TCT CCC CTT TCC TGG TTG TC	CCA GTA GGC GAG CTT CTT CTT CA	65
Panx1	CAC CGA GCC CAA GTT CAA GG	GGC CCA GCA GTA AGA GTC CA	64
Il1b	TCT GTG ACT CGT GGG ATG AT	CAC TTG TTG GCT TAT GTT CTG TC	61
Tnf	GTC CAA CTC CGG GCT CAG AAT	ACT CCC CCG ATC CAC TCA G	65
Il6	GCC ACT GCC TTC CCT ACT TCA C	GAC AGT GCA TCA TCG CTG TTC ATA C	63
Tgfb	GCG CCT GCA GAG ATT CAA GTC AAC	TCA GGC GTA TCA GTG GGG GTC A	65
Ywhaz	TTG AGC AGA AGA CGG AAG GT	GAA GCA TTG GGG ATC AAG AA	63
Osbp	TCC GGG AGA CTT TAC CTT CAC TT	GTG TCA CCC TCT TAT CAA CCA CC	63
Hprt1	CGT CGT GAT TAG TGA TGA TGA AC	CAA GTC TTT CAG TCC TGT CCA TAA	65

## Data Availability

Data is contained within the article.

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
