# Peer review of "A Single Episode of Cortical Spreading Depolarization Increases mRNA Levels of Proinflammatory Cytokines, Calcitonin Gene-Related Peptide and Pannexin-1 Channels in the Cerebral Cortex"

_ijms, 2022, doi:10.3390/ijms24010085_

Round 1
Reviewer 1 Report
This paper demonstrated that a single CSD induces neuroinflammation 3 hours post CSD, to a lower extent, in the intact cortex than in the injury cortex of rat, which is very interesting. The authors monitored CSD and mRNA levels of selected genes. As the associated protein expression data were not presented, the title ‘ A single episode of cortical spreading depolarization causes 2 low grade neuroinflammation and overexpression of calcitonin 3 gene-related peptide and pannexin-1 channels in the cerebral 4 cortex’ is thus somewhat overstated. There are some spelling mistakes and inconsistent symbols are to corrected. Detailed comments are shown below.
Line 129 different fonts are seen, suggest to keep consistent.
Line 150: Suggest to add direction of primer sequences
Line 162: Missing letter ‘Fi 1’
Line 170: TNF, TNF-alpha, IL-1b, IL1b are seen. Please check to keep spelling for inflammatory factors being kept consistent.
Line 260: ‘Our data about a lack of significant 260 changes in CGRP expression in the perilesioned cortical tissue indicate that CSD is more 261 important factor for CGRP upregulation that acute parenchymal injury.’ Shoud ‘that’ be ‘than’?
Line243-244: The authors stated that ‘The current study for the first time shows persistent overexpression of Pannexin-1 (Panx1) channels following a single CSD. ‘ This is overstated as the author did not present Panx1 protein expression data besides gene expression. Suggest to revise throughout as similar statement based on mRNA levels of Il1b, TNF, Il6 are also seen.
In the same paragraph, it is stated that ‘Therefore, upregulation of Panx1 channels following a single CSD event found in our study may lead to intensification of cytokine production and cortical neuroinflammatory response to subsequent repeated CSDs that may contribute to sensitization process and migraine chronification.’ Does the authors meant to suggest a causal relationship between Panx1 and neuroinflammatory response? Based on the data presented, upregulation of Panx1 and TNF-alpha gene expression at the same time point post CSD may be just parallel event. Please explain.
It will be interesting to explain the possible mechanism underlying upregulation of gene expression of Panx1, CGRP and TNF-alpha in intact anterior region of the cortex post CSD.
The authors induced CSD in the left cerebral cortex and sham group on in the right, then compare CSD and mRNA levels between the two sides. Is there any possible influence on these parameters by cortical structure difference between the two sides?
Author Response
We thank you very much for the valuable comments and suggestions. Our point-to point response follows.
Point 1. This paper demonstrated that a single CSD induces neuroinflammation 3 hours post CSD, to a lower extent, in the intact cortex than in the injury cortex of rat, which is very interesting. The authors monitored CSD and mRNA levels of selected genes. As the associated protein expression data were not presented, the title ‘A single episode of cortical spreading depolarization causes low grade neuroinflammation and overexpression of calcitonin gene-related peptide and pannexin-1 channels in the cerebral cortex’ is thus somewhat overstated.
RESPONSE: We agree with the comment and changed the title to the next “A single episode of cortical spreading depolarization increases mRNA expression of pro-inflammatory cytokines, calcitonin gene-related peptide and pannexin-1 channels in the cerebral cortex”
Point 2. Line 129 different fonts are seen, suggest to keep consistent.
RESPONSE: Done.
Point 3. Line 150: Suggest to add direction of primer sequences
RESPONSE: Done (line 170)
Point 4. Line 162: Missing letter ‘Fi 1’
RESPONSE: Done (line 182).
Point 5. Line 170: TNF, TNF-alpha, IL-1b, IL1b are seen. Please check to keep spelling for inflammatory factors being kept consistent.
RESPONSE: Sorry, the spelling of cytokines in the text of the manuscript differed from that used in Figures 3-4. Now we corrected the abbreviations on the figures in accordance with the text.
Point 6. Line 260: ‘Our data about a lack of significant changes in CGRP expression in the perilesioned cortical tissue indicate that CSD is more important factor for CGRP upregulation that acute parenchymal injury.’ Should ‘that’ be ‘than’?
RESPONSE: Done (line 298).
Point 7. Line243-244: The authors stated that ‘The current study for the first time shows persistent overexpression of Pannexin-1 (Panx1) channels following a single CSD.‘ This is overstated as the author did not present Panx1 protein expression data besides gene expression. Suggest to revise throughout as similar statement based on mRNA levels of Il1b, TNF, Il6 are also seen.
RESPONSE: Agree. We corrected this statement and others on lines 293, 274, 250, 241, 210, 190, 90.
Point 8. In the same paragraph, it is stated that ‘Therefore, upregulation of Panx1 channels following a single CSD event found in our study may lead to intensification of cytokine production and cortical neuroinflammatory response to subsequent repeated CSDs that may contribute to sensitization process and migraine chronification.’ Does the authors meant to suggest a causal relationship between Panx1 and neuroinflammatory response? Based on the data presented, upregulation of Panx1 and TNF-alpha gene expression at the same time point post CSD may be just parallel event. Please explain.
RESPONSE: A causative relationship between Panx1 activation and neuroinflammation following CSD has been shown by Karatas et al. (2013) cited in our study [14]. Now we provided additional details on the relationship in Discussion (lines 281-285): “It has been demonstrated that rapid activation of Panx1 within minutes following CSD triggers inflammasome formation and the release of pro-inflammatory molecules such as Il1b. Inhibitors of Panx1 channels suppress the inflammatory signaling cascade and abolish CSD-induced trigeminal activation [14].”
Point 9. It will be interesting to explain the possible mechanism underlying upregulation of gene expression of Panx1, CGRP and TNF-alpha in intact anterior region of the cortex post CSD.
RESPONSE: There are multiple mechanisms by which CSD can affect gene expression in the intact cortical tissue and of course the suggestions may be only speculative. It is known that strong membrane depolarization during CSD evokes the robust calcium influx into the cytoplasm. Given that calcium signaling is a potent trigger of gene expression, we think that the increase in intracellular calcium concentrations during CSD activates the transcriptional response. We have included the suggestion in Discussion (lines 321-323): “Mechanisms underlying CSD-induced upregulation of gene expression may involve calcium signaling [40] activated by the calcium influx into the cytoplasm during CSD although other mechanisms cannot be excluded.”
Point 10. The authors induced CSD in the left cerebral cortex and sham group on in the right, then compare CSD and mRNA levels between the two sides. Is there any possible influence on these parameters by cortical structure difference between the two sides?
RESPONSE: No evidence for interhemispheric difference in cortical structure has been found in the rat brain. The design of our experiments was aimed at minimization of the animal use and represents the common approach to studying molecular effects of unilaterally induced CSD (e.g. see Takizawa et al., 2019).
We thank you very much for the valuable comments and suggestions. Our point-to point response follows.
Point 1. This paper demonstrated that a single CSD induces neuroinflammation 3 hours post CSD, to a lower extent, in the intact cortex than in the injury cortex of rat, which is very interesting. The authors monitored CSD and mRNA levels of selected genes. As the associated protein expression data were not presented, the title ‘A single episode of cortical spreading depolarization causes low grade neuroinflammation and overexpression of calcitonin gene-related peptide and pannexin-1 channels in the cerebral cortex’ is thus somewhat overstated.
RESPONSE: We agree with the comment and changed the title to the next “A single episode of cortical spreading depolarization increases mRNA expression of pro-inflammatory cytokines, calcitonin gene-related peptide and pannexin-1 channels in the cerebral cortex”
Point 2. Line 129 different fonts are seen, suggest to keep consistent.
RESPONSE: Done.
Point 3. Line 150: Suggest to add direction of primer sequences
RESPONSE: Done (line 170)
Point 4. Line 162: Missing letter ‘Fi 1’
RESPONSE: Done (line 182).
Point 5. Line 170: TNF, TNF-alpha, IL-1b, IL1b are seen. Please check to keep spelling for inflammatory factors being kept consistent.
RESPONSE: Sorry, the spelling of cytokines in the text of the manuscript differed from that used in Figures 3-4. Now we corrected the abbreviations on the figures in accordance with the text.
Point 6. Line 260: ‘Our data about a lack of significant changes in CGRP expression in the perilesioned cortical tissue indicate that CSD is more important factor for CGRP upregulation that acute parenchymal injury.’ Should ‘that’ be ‘than’?
RESPONSE: Done (line 298).
Point 7. Line243-244: The authors stated that ‘The current study for the first time shows persistent overexpression of Pannexin-1 (Panx1) channels following a single CSD.‘ This is overstated as the author did not present Panx1 protein expression data besides gene expression. Suggest to revise throughout as similar statement based on mRNA levels of Il1b, TNF, Il6 are also seen.
RESPONSE: Agree. We corrected this statement and others on lines 293, 274, 250, 241, 210, 190, 90.
Point 8. In the same paragraph, it is stated that ‘Therefore, upregulation of Panx1 channels following a single CSD event found in our study may lead to intensification of cytokine production and cortical neuroinflammatory response to subsequent repeated CSDs that may contribute to sensitization process and migraine chronification.’ Does the authors meant to suggest a causal relationship between Panx1 and neuroinflammatory response? Based on the data presented, upregulation of Panx1 and TNF-alpha gene expression at the same time point post CSD may be just parallel event. Please explain.
RESPONSE: A causative relationship between Panx1 activation and neuroinflammation following CSD has been shown by Karatas et al. (2013) cited in our study [14]. Now we provided additional details on the relationship in Discussion (lines 281-285): “It has been demonstrated that rapid activation of Panx1 within minutes following CSD triggers inflammasome formation and the release of pro-inflammatory molecules such as Il1b. Inhibitors of Panx1 channels suppress the inflammatory signaling cascade and abolish CSD-induced trigeminal activation [14].”
Point 9. It will be interesting to explain the possible mechanism underlying upregulation of gene expression of Panx1, CGRP and TNF-alpha in intact anterior region of the cortex post CSD.
RESPONSE: There are multiple mechanisms by which CSD can affect gene expression in the intact cortical tissue and of course the suggestions may be only speculative. It is known that strong membrane depolarization during CSD evokes the robust calcium influx into the cytoplasm. Given that calcium signaling is a potent trigger of gene expression, we think that the increase in intracellular calcium concentrations during CSD activates the transcriptional response. We have included the suggestion in Discussion (lines 321-323): “Mechanisms underlying CSD-induced upregulation of gene expression may involve calcium signaling [40] activated by the calcium influx into the cytoplasm during CSD although other mechanisms cannot be excluded.”
Point 10. The authors induced CSD in the left cerebral cortex and sham group on in the right, then compare CSD and mRNA levels between the two sides. Is there any possible influence on these parameters by cortical structure difference between the two sides?
RESPONSE: No evidence for interhemispheric difference in cortical structure has been found in the rat brain. The design of our experiments was aimed at minimization of the animal use and represents the common approach to studying molecular effects of unilaterally induced CSD (e.g. see Takizawa et al., 2019).
We thank you very much for the valuable comments and suggestions. Our point-to point response follows.
Point 1. This paper demonstrated that a single CSD induces neuroinflammation 3 hours post CSD, to a lower extent, in the intact cortex than in the injury cortex of rat, which is very interesting. The authors monitored CSD and mRNA levels of selected genes. As the associated protein expression data were not presented, the title ‘A single episode of cortical spreading depolarization causes low grade neuroinflammation and overexpression of calcitonin gene-related peptide and pannexin-1 channels in the cerebral cortex’ is thus somewhat overstated.
RESPONSE: We agree with the comment and changed the title to the next “A single episode of cortical spreading depolarization increases mRNA expression of pro-inflammatory cytokines, calcitonin gene-related peptide and pannexin-1 channels in the cerebral cortex”
Point 2. Line 129 different fonts are seen, suggest to keep consistent.
RESPONSE: Done.
Point 3. Line 150: Suggest to add direction of primer sequences
RESPONSE: Done (line 170)
Point 4. Line 162: Missing letter ‘Fi 1’
RESPONSE: Done (line 182).
Point 5. Line 170: TNF, TNF-alpha, IL-1b, IL1b are seen. Please check to keep spelling for inflammatory factors being kept consistent.
RESPONSE: Sorry, the spelling of cytokines in the text of the manuscript differed from that used in Figures 3-4. Now we corrected the abbreviations on the figures in accordance with the text.
Point 6. Line 260: ‘Our data about a lack of significant changes in CGRP expression in the perilesioned cortical tissue indicate that CSD is more important factor for CGRP upregulation that acute parenchymal injury.’ Should ‘that’ be ‘than’?
RESPONSE: Done (line 298).
Point 7. Line243-244: The authors stated that ‘The current study for the first time shows persistent overexpression of Pannexin-1 (Panx1) channels following a single CSD.‘ This is overstated as the author did not present Panx1 protein expression data besides gene expression. Suggest to revise throughout as similar statement based on mRNA levels of Il1b, TNF, Il6 are also seen.
RESPONSE: Agree. We corrected this statement and others on lines 293, 274, 250, 241, 210, 190, 90.
Point 8. In the same paragraph, it is stated that ‘Therefore, upregulation of Panx1 channels following a single CSD event found in our study may lead to intensification of cytokine production and cortical neuroinflammatory response to subsequent repeated CSDs that may contribute to sensitization process and migraine chronification.’ Does the authors meant to suggest a causal relationship between Panx1 and neuroinflammatory response? Based on the data presented, upregulation of Panx1 and TNF-alpha gene expression at the same time point post CSD may be just parallel event. Please explain.
RESPONSE: A causative relationship between Panx1 activation and neuroinflammation following CSD has been shown by Karatas et al. (2013) cited in our study [14]. Now we provided additional details on the relationship in Discussion (lines 281-285): “It has been demonstrated that rapid activation of Panx1 within minutes following CSD triggers inflammasome formation and the release of pro-inflammatory molecules such as Il1b. Inhibitors of Panx1 channels suppress the inflammatory signaling cascade and abolish CSD-induced trigeminal activation [14].”
Point 9. It will be interesting to explain the possible mechanism underlying upregulation of gene expression of Panx1, CGRP and TNF-alpha in intact anterior region of the cortex post CSD.
RESPONSE: There are multiple mechanisms by which CSD can affect gene expression in the intact cortical tissue and of course the suggestions may be only speculative. It is known that strong membrane depolarization during CSD evokes the robust calcium influx into the cytoplasm. Given that calcium signaling is a potent trigger of gene expression, we think that the increase in intracellular calcium concentrations during CSD activates the transcriptional response. We have included the suggestion in Discussion (lines 321-323): “Mechanisms underlying CSD-induced upregulation of gene expression may involve calcium signaling [40] activated by the calcium influx into the cytoplasm during CSD although other mechanisms cannot be excluded.”
Point 10. The authors induced CSD in the left cerebral cortex and sham group on in the right, then compare CSD and mRNA levels between the two sides. Is there any possible influence on these parameters by cortical structure difference between the two sides?
RESPONSE: No evidence for interhemispheric difference in cortical structure has been found in the rat brain. The design of our experiments was aimed at minimization of the animal use and represents the common approach to studying molecular effects of unilaterally induced CSD (e.g. see Takizawa et al., 2019).

Author Response
We thank you very much for the valuable comments and suggestions. Our point-to point response follows.
Point 1. Authors should add some further details about the way in which anesthesia can modify an innate immune response of the injured brain by adding other references in particular about potential mechanisms by which the immune system affects the central nervous system.
RESPONSE: Thank you very much for the suggestions. We have added the details on the effects of anesthesia in the injured brain on lines 64-68 (“General anesthesia can modify an innate immune response of the injured brain [16, 17] and produce long-term cognitive impairments [18]. Patterns and effects of CSD are known to depend on the baseline functional state of the brain. Significant changes in the state under general anesthesia suggest that characteristics of neuroinflammatory responses may differ in awake and anesthetized animals.”) and lines 35-44 (“The interplay between immune and neuronal systems is involved in regulation of physiological functions of the brain and pathogenic mechanisms of many neurological conditions. Innate immune cells of brain parenchyma activated by exogeneous or endogeneous stress signals initiate an orchestrated neuroinflammatory response with enhanced production of cytokines [1, 2]. The immune response of the brain may be triggered by enhanced neuronal activity in the absence of significant pathological conditions (neurogenic neuroinflammation) [2]. Neuroinflammation is known to produce both beneficial (tissue remodeling, repair, coping with enhanced metabolic demands etc.) and detrimental (pain promoting, sensitization, neurotoxicity etc.) effects on the brain function [1].”).
Point 2. Moreover, they should add some notes concerning the genetic basis of migraine so that this study could offer interesting insights about genetic aspects of the observed results.
RESPONSE: Respective sentence has been added: “Migraine has a strong genetic component and genetic factors are likely to play a role in individual susceptibility to CSD [3].” (lines 49-50).
Point 3. In particular they should cite these references: • Irene Simonetta, Renata Riolo, Federica Todaro, Antonino Tuttolomondo. New Insights on Metabolic and Genetic Basis of Migraine: Novel Impact on Management and Therapeutical Approach. Int J Mol Sci. 2022 Mar 11;23(6):3018. doi: 10.3390/ijms23063018. • Rasmussen, L. S. et al. Does anaesthesia cause postoperative cognitive dysfunction? A randomised study of regional versus general anaesthesia in 438 elderly patients. Acta Anaesthesiol. Scand. 47, 260–266 (2003). Inouye, S. K., Westendorp, R. G. J. & Saczynski, J. S. Delirium in elderly • Huber-Lang, M., Lambris, J. D. & Ward, P. A. Innate immune responses to trauma. Nat. Immunol. 19, 327–341 (2018).
RESPONSE: Thank you for the suggestions. All of these references and several additional papers are cited in the revised version of the manuscript.
Point 4. In my opinion, authors should argue the role of Panx1 in the paragraph between lines 250-256.
RESPONSE: The discussion of the role of Panx1 has been expanded (lines 281-285).
Point 5. Finally, I think that authors should report the duration of the effects occurring after a single CSD; it could be useful in order to better understand the influence of a single CSD in the pathogenesis of migraine. These aspects could complete and make more satisfactory their original study.
RESPONSE: In the revised version of the manuscript, the duration of the observed effects is discussed on lines 255-262: “Most CSD-associated events (breakdown of transmembrane ion gradients, glutamate release, cellular depolarization etc.) last minutes in well-nourished neuronal tissue, but the relatively brief and reversible dysfunction is followed by sustained upregulation of pro-inflammatory cytokines and nociceptive mediators in the cerebral cortex. Given the timing of the aura phase relative to migraine headache, the activation of the parenchymal inflammatory and nociceptive pathways for hours following a single CSD episode supports a potential involvement of CSD in mechanisms of pain generation in migraine with aura.”

Round 2
Reviewer 1 Report
The authors addressed majority questions and improved the paper quality and corrected format and grammars. In the meanwhile, the following points seem to be insufficiently addressed. They are:
Point 1. mRNA expression is confusion as protein level was not presented. Suggest changing 'mRNA expression' to mRNA level or gene expression.
Point 5. Change IL-1b to IL-1beta throughout
Point 9. The authors added discussion that “Mechanisms underlying CSD-induced upregulation of gene expression may involve calcium signaling [40] activated by the calcium influx into the cytoplasm during CSD although other mechanisms cannot be excluded.” in an attempt to explain 'possible mechanism underlying upregulation of gene expression of Panx1, CGRP and TNF-alpha in intact anterior region of the cortex post CSD. ' This explanation is insufficient. Upregulation of Panx1, CGRP and TNF-alpha mRNA gene expression by CSD is the major finding of the study, so the underlying mechanism is expected to be addressed in sufficient details. One possibility is that NFkB is important for inflammatory responses (Liu, T et al. NF-kB signaling in inflammation, Signal Transduct. 2017). Could this transcription factor be recruited to promote cytokine gene expression, especially IL-1beta? Alternative mechanisms may exist for different genes concerned.
Point 10. Comments. This is understandable. Perhaps, the authors could design by inducing 50% CSD in the right hemispheres and another 50% in the left hemisphere, this will help to show if there would be interhemispheric difference in specific gene expression post CSD.
Author Response
Point 1. mRNA expression is confusion as protein level was not presented. Suggest changing 'mRNA expression' to mRNA level or gene expression.
RESPONSE: We think that “gene expression” may be confused because the term includes both mRNA and protein expression (e.g. see Buccitelli C, Selbach M. mRNAs, proteins and the emerging principles of gene expression control. Nat Rev Genet. 2020 Oct;21(10):630-644. doi: 10.1038/s41576-020-0258-4). Addressing the comment, we have changed the title to the next “A single episode of cortical spreading depolarization increases mRNA levels of pro-inflammatory cytokines, calcitonin gene-related peptide and pannexin-1 channels in the cerebral cortex”
Point 5. Change IL-1b to IL-1beta throughout.
RESPONSE: Il1b is the official symbol of the gene (https://www.ncbi.nlm.nih.gov/gene/24494). In the keywords, we provide its official full name - interleukin-1beta.
Point 9. The authors added discussion that “Mechanisms underlying CSD-induced upregulation of gene expression may involve calcium signaling [40] activated by the calcium influx into the cytoplasm during CSD although other mechanisms cannot be excluded.” in an attempt to explain 'possible mechanism underlying upregulation of gene expression of Panx1, CGRP and TNF-alpha in intact anterior region of the cortex post CSD. ' This explanation is insufficient. Upregulation of Panx1, CGRP and TNF-alpha mRNA gene expression by CSD is the major finding of the study, so the underlying mechanism is expected to be addressed in sufficient details. One possibility is that NFkB is important for inflammatory responses (Liu, T et al. NF-kB signaling in inflammation, Signal Transduct. 2017). Could this transcription factor be recruited to promote cytokine gene expression, especially IL-1beta? Alternative mechanisms may exist for different genes concerned.
RESPONSE: The interest of Reviewer to mechanisms underlying the observed effects of CSD on gene expression is understandable. But investigation of the mechanisms was out of the scope of our study. Our results do not allow making any definite conclusions about the point and we do not want to multiplicate speculative ungrounded suggestions in Discussion.
Point 10. This is understandable. Perhaps, the authors could design by inducing 50% CSD in the right hemispheres and another 50% in the left hemisphere, this will help to show if there would be interhemispheric difference in specific gene expression post CSD.
RESPONSE: We will consider the advice in our future experiments.